# TERT Promoter Methylation Is Oxygen-Sensitive and Regulates Telomerase Activity

**DOI:** 10.3390/biom14010131

**Published:** 2024-01-19

**Authors:** Fatma Dogan, Nicholas R. Forsyth

**Affiliations:** 1Vaccine and Immunotherapy Center, Massachusetts General Hospital, Harvard Medical School, Boston, MA 02129, USA; 2The Guy Hilton Research Laboratories, School of Pharmacy and Bioengineering, Faculty of Medicine and Health Sciences, Keele University, Stoke on Trent ST4 7QB, UK; 3Vice Principals Office, Kings College, University of Aberdeen, Aberdeen AB24 3FX, UK

**Keywords:** telomerase, TERT promoter, DNMT3B, pluripotent stem cells, characterization, epigenetic, methylation, physiological oxygen, DNA methyltransferase

## Abstract

Telomere repeats protect linear chromosomes from degradation, and telomerase has a prominent role in their maintenance. Telomerase has telomere-independent effects on cell proliferation, DNA replication, differentiation, and tumorigenesis. TERT (telomerase reverse transcriptase enzyme), the catalytic subunit of telomerase, is required for enzyme activity. TERT promoter mutation and methylation are strongly associated with increased telomerase activation in cancer cells. TERT levels and telomerase activity are downregulated in stem cells during differentiation. The link between differentiation and telomerase can provide a valuable tool for the study of the epigenetic regulation of TERT. Oxygen levels can affect cellular behaviors including proliferation, metabolic activity, stemness, and differentiation. The role of oxygen in driving TERT promoter modifications in embryonic stem cells (ESCs) is poorly understood. We adopted a monolayer ESC differentiation model to explore the role of physiological oxygen (physoxia) in the epigenetic regulation of telomerase and TERT. We further hypothesized that DNMTs played a role in physoxia-driven epigenetic modification. ESCs were cultured in either air or a 2% O_2_ environment. Physoxia culture increased the proliferation rate and stemness of the ESCs and induced a slower onset of differentiation than in ambient air. As anticipated, downregulated TERT expression correlated with reduced telomerase activity during differentiation. Consistent with the slower onset of differentiation in physoxia, the TERT expression and telomerase activity were elevated in comparison to the air-oxygen-cultured ESCs. The TERT promoter methylation levels increased during differentiation in ambient air to a greater extent than in physoxia. The chemical inhibition of DNMT3B reduced TERT promoter methylation and was associated with increased TERT gene and telomerase activity during differentiation. DNMT3B ChIP (Chromatin immunoprecipitation) demonstrated that downregulated TERT expression and increased proximal promoter methylation were associated with DNMT3B promoter binding. In conclusion, we have demonstrated that DNMT3B directly associates with TERT promoter, is associated with differentiation-linked TERT downregulation, and displays oxygen sensitivity. Taken together, these findings help identify novel aspects of telomerase regulation that may play a role in better understanding developmental regulation and potential targets for therapeutic intervention.

## 1. Introduction

Embryonic stem cells (ESCs) have inherent properties including self-renewal, immortality, pluripotency, endogenous telomerase activity, and long telomeres [1,2]. The pluripotency attribute of ESCs enables their differentiation into cell types representative of the endoderm, ectoderm, and mesoderm germ layers [3]. The enzyme telomerase is highly expressed in ESCs, and through the maintenance of telomere length, plays a role in maintaining pluripotency, immortality, and self-renewal [4,5]. Somatic cells express very low or no detectable telomerase [6].

The telomerase enzyme consists of essential protein and RNA (TERC) components and is the primary mechanism for telomere maintenance and elongation [7,8]. TERT, the catalytic protein subunit, is a rate-limiting factor for telomerase activity [9]. The knockdown of TERT results in the loss of pluripotency, loss of clonogenicity, and spontaneous differentiation of ESCs [10]. TERT transcriptional silencing during the differentiation of stem cells and its activation during the transformation of somatic into cancer cells remains incompletely understood. It is, however, clear that genetic and epigenetic changes play a role in the regulation of the TERT gene, including promoter mutation and methylation, histone modification, and non-coding RNAs [11,12]. 

Epigenetic modifications have a crucial role in cell fate determination including the upregulation of lineage-specific genes, decreased expression of self-renewal related genes [13], and decreased TERT expression and telomerase enzyme activity during differentiation [14,15,16,17]. Undifferentiated ESCs have a unique epigenetic signature in comparison to differentiated ESCs, along with somatic cells [18,19,20,21]. DNMT3A and DNMT3B are highly expressed in undifferentiated ESCs, and downregulated during differentiation [22,23,24]. DNMT1, DNMT3A, and DNMT3B are the three major DNA methyltransferases responsible for methylation in mammals [25,26]. DNMT1 can distinguish hemimethylated DNA during DNA replication and acts to maintain global methylation patterns following replication [27]. DNMT3A and DNMT3B de nova methyltransferases establish DNA methylation patterns during gametogenesis, embryogenesis, and somatic tissue development, working in coordination with DNMT1 [28]. DNMT3L (DNA methyltransferase 3-like) cooperates with DNMT3A and DNMT3B to stimulate their catalytic activity and is also highly expressed in ESCs [29]. Somatic cells and ESCs display distinct DNA methylation signatures associated with lineage specification [21,30,31]. The chemical inhibition of DNMT3B with adriamycin and azacytidine reduced human TERT (hTERT) expression and, in changing GC to AT, abolished promoter activity through site-directed mutagenesis in glioma cell lines, demonstrating that DNMT3B and GC islands in the TERT promoter play an important role in the regulation of telomerase expression [12,32]. 

Physiological oxygen levels vary across organs and tissue compartments and typically range from 1 to 14% in vivo for most tissues depending on the distance away from the vascular system [33,34,35]. The physiological normoxia (physoxia) environment for ESCs in their pre-isolation setting typically ranges from 2 to 5% [36]. Oxygen affects epigenetic modifications, and epigenetics is essential for the initiation of hypoxic response pathways [37]. Cells cultured in air oxygen conditions have higher oxidative stress, DNA damage, genomic instability, and stress-linked senescence due to the formation of reactive oxygen species [34]. Reduced oxygen tension-based culture can alter stem cell characteristics including proliferation, differentiation, pluripotency, genomic stability, and DNA methylation [11,38,39,40,41]. Further, low-oxygen, physoxic culture conditions decrease global DNA methylation levels across a range of cancer and stem cells [42,43,44]. Therefore, investigating the role of physoxia on discrete epigenetic profiles is increasingly relevant for mammalian stem cell culture. We hypothesized that the oxygen-linked methylation of the TERT promoter would play a role in the modulation and discrete control of the telomerase enzyme activity during differentiation.

Here, we have investigated the effect of physoxia on ESC differentiation and telomerase activity. We have explored the association between TERT expression and gene promoter methylation and the role that DNMT3B has to play. Consistent with previous reports, we noted a higher proliferation rate of ESCs in reduced oxygen culture. Monolayer differentiated cells displayed the upregulation of mesodermal markers predominantly associated with a slower onset of differentiation in physoxia. The telomerase activity, TERT expression, and telomere length decreased during differentiation correlated to the increased methylation on the TERT promoter, all to a lesser extent in physoxia. The ChIP demonstrated that DNMT3B binding to the promoter region is associated with the TERT expression levels and increased proximal methylation. Our data suggested that methylation, oxygen environment, and DNMTs play a role in TERT regulation during hESC (Human embryonic stem cell) differentiation.

## 2. Materials and Methods

### 2.1. Cell Culture

hESC lines (SHEF1 and SHEF2) were used under approval from the UK Stem Cell Bank (UKSCB) [45]. SHEF1 and SHEF2 cells were cultured in E8 medium (Life Technologies, London, UK) with Essential 8 Supplement (1X) in culture vessels coated with 5 mg/mL vitronectin (Recombinant Human Protein; Life Technologies, London, UK). Spontaneous differentiation medium comprised Knockout DMEM (Thermofisher, Manchester, UK), 10% FBS, 1% NEAA, 1% L-glutamine and β-mercaptoethanol. hESCs were grown in 6-well vitronectin-coated culture plates in three conditions for 48 h in E8 medium before switching into spontaneous differentiation media. Cells were maintained in either air oxygen (21% AO), a fully defined 2% O_2_ environment (workstation) (2% WKS), or a standard 2% O_2_ incubator (2% PG), where samples were handled in a standard class II biological safety laminar flow cabinet. Media were deoxygenated to a 2% O_2_ level using defined Hypoxycool (Baker Ruskinn, Bridgend, UK) cycle settings. Pre-gassed media (pre-conditioned to 2% O_2_ in a HypoxyCool unit) was used in all 2% O2 experimentation. Cell proliferation was assessed using WST1 assay (Sigma-Aldrich, Dorset, UK) (Appendix A). Nanaomycin A (NA) dose was determined with WST1 to establish maximum non-toxic concentrations. The DNMT3B selective inhibitor (nanaomycin A) was purchased from Adooq Bioscience (Irvine, CA, USA) and dissolved in DMSO. Cell viability was determined with serial dilution of drugs ranging from 20 μM to 80 nM over seven days using WST1 assay. 

### 2.2. Characterization of ESCs

Undifferentiated hESCs were characterized for retention of pluripotency marker expression during differentiation using the Human Pluripotent Stem Cell Marker Antibody Panel (R&D systems, London, UK) at days 5, 10, 20, and 40 (Appendix A). 

### 2.3. Gene Expression

RNA was isolated from hESCs with the RNeasy^®^ Mini Kit (Qiagen, Manchester, UK), and concentration quantified using a NanoDropTM 2000/2000c Spectrophotometer (Thermo Scientific, Cambridge, UK). qRT-PCR was performed using the QuantiFast SYBR Green OneStep qRT-PCR kit (Qiagen, Manchester, UK). PCR primers were from Sigma-Aldrich, UK. Primer sequences and product sizes are listed in Appendix A. OneStep qRT-PCR was performed with 25 ng of input RNA, and relative quantification of gene expression was measured using the −ΔΔCT method. 

### 2.4. Telomerase Activity 

The TRAPeze^®^ RT Telomerase Detection Kit (Millipore, Billerica, MA, USA) was applied with Amplifluor^®^ primers to detect telomerase activity. Protein extracts from cell pellets were prepared using CHAPS Lysis Buffer and stored at −75 °C to −85 °C. qRT-PCR followed manufacturer’s protocol. The standard curve of the control template was used to quantify telomerase activity via fluorometric detection. 

### 2.5. Telomere Length Quantification 

Absolute Human Telomere Length Quantification qPCR assay (Caltag Medsystems Limited, Buckingham, UK) was used to measure telomere length. A single-copy reference primer set recognizes a 100 bp long region on human chromosome 17 and is used as a reference for each sample. A reference genomic DNA sample provided a standard of data normalization with known telomere length for calculation of telomere length of target samples. DNA extraction (Qiagen, UK) was performed for each cell pellet. qRT-PCR followed manufacturer protocol. Data from qRT-PCR were analyzed according to manufacturer calculations.

### 2.6. Pyrosequencing

EZ DNA Methylation-Gold™ Kit (Zymo Research, Orange, CA, USA) was used for bisulphite conversion of 500 ng genomic DNA. TERT gene promoter region sequences were designed via the PyroMark Q24 Software 2.0. Primer sequences, locations, and expected PCR product size are listed in Appendix A and supplied by Biomers (Ulm, Germany).

Converted DNA (2–4 μL) was utilized as a template for PCR amplification. PCR reactions were performed with GoTaq^®^G2 Flexi DNA Polymerase kit (Promega, Southampton, UK). Initial denaturation was achieved by cycling at 95 °C for 5 min, followed by touch-down cycling for the first 14 cycles, with the temperature decreasing by 0.5 °C each cycle. Then, 35 cycles of 95 °C for 45 s, annealing at 55–63 °C for 45 s, elongation at 72 °C for 30 s, and a final elongation step of 72 °C for 5 min were performed. The quality of the PCR amplification product was validated using a 2% agarose gel electrophoresis. Combination of PCR products was performed with streptavidin-sepharose beads that enabled biotin-labeled PCR amplicon collection (GE Healthcare, Chicago, IL, USA). Following purification, beads with biotin-labeled PCR amplicons were released into an annealing mix. PyroMark Gold Q24 Reagents, including the four nucleotides, the substrate, and the enzyme mix, were placed into a pyrosequencing dispensation cartridge which was inserted, along with the Q24 pyrosequencing plate, into the pyrosequencing device. Data analysis was performed with PyroMark Q24 Software 2.0.

### 2.7. DNMT3B Activity Assay

EpiQuik™ DNA Methyltransferase 3B Activity Assay (Insight Biotechnology, Wembley, UK) was used to analyze DNMT3B activity. Briefly, a 3 μL (10 μg) sample was introduced into individual wells followed by 27 μL of DNMT Assay Buffer and 3 μL of diluted Adomet (8 mM), and then incubated at 37 °C for 90 min. Wells were then washed three times with 150 μL 1X Wash Buffer. Each well was filled with diluted Capture Antibody solution (50 μL), and then incubated at room temperature for 60 min on an orbital shaker (50–100 rpm). Wells were then aspirated and washed three times with 150 μL 1X Wash Buffer. Detection antibody (50 µL) was added to each well followed by incubation at room temperature for 30 min. Wells were then washed four times with 150 µL 1X Wash Buffer. Enhancer solution (50 µL) was added to each well followed by incubation at room temperature for 30 min. The wells then received their final wash, four times, with 150 µL 1X Wash Buffer. Developing solution (100 µL) was added to each well and then incubated at room temperature for 2–10 min in the dark. Stop solution (50 µL) was added and then absorbance read on a microplate reader at 450 nm within 5–15 min.

### 2.8. DNMT3B-TERT Promoter Chromatin Immunoprecipitation (ChIP)

ChIP-IT^®^ Express Chromatin Immunoprecipitation Kit (Active Motif, Rosemont, IL, USA) was used to investigate the binding of DNMT3B antibody to TERT promoter regions. hESC pellets (1.5 × 10^7^) were first fixed with formaldehyde to cross-link and stabilize protein/DNA complexes. The cell pellet was resuspended in 1 mL cold lysis buffer supplemented with 5 µL PMSF (phenylmethylsulfonyl fluoride, a serine protease inhibitor) and 1 µL Protease Inhibitor Cocktail (PIC). The mixture was vortexed to mix and then incubated on ice for 30 min. To pellet the nuclei, samples were centrifuged for 10 min at 5000 rpm at 4 °C, and the supernatant carefully discarded. The nuclei pellet was resuspended in 350 µL digestion buffer with 1.75 µL PIC and 1.75 µL PMSF and incubated for 5 min at 37 °C. Enzymatic shearing cocktail was added to the nuclei and incubated for 5 min at 37 °C. The reaction was stopped via addition of 7 µL cold 0.5 M EDTA and incubation for 10 min on ice. ChIP DNA Purification Kit (Active Motif, USA) was used to clean up 50 µL DNA samples and establish concentration.

DNA fragments with specific DNMT3B protein interaction were captured with DNMT3B antibody and protein-G coated magnetic beads. The ChIP-IT^®^ Control Kit (Active Motif, USA) was used to assess the non-specific binding of the DNMT3B antibody. A total of 25 µg of DNA was used for ChIP reactions. ChIP reactions were prepared using 25 µL Protein-G coated magnetic beads, 10 µL ChIP buffer 1, 20–60 µL sheared chromatin (25 µg), 1 µL PIC and dH_2_O to complete the final volume of 100 µL and 2 µg DNMT3B (mouse IgG) antibody. ChIP reaction positive (2 µg RNA pol II and 2 µg bridging antibody) and negative control (IgG antibody) were included: 25 µL Protein-G coated magnetic beads, 10 µL ChIP buffer 1, 20–60 µL sheared chromatin (25 µg), 1 µL PIC and dH_2_O to complete the final volume to 100 µL. Following overnight incubation, samples were spun briefly to collect, and placed on the magnetic stand to pellet the magnetic beads. After washing and reverse cross-linking, DNA was eluted from the beads. Proteins were removed, and DNA purified with ChIP DNA Purification Kit for downstream analysis. Primers for TERT promoter regions for ChIP samples were designed and are listed in Appendix A. qPCR reactions were performed on samples using the SYBR Green Master Mix (Thermofisher, UK). Serial dilutions of input DNA for each primer set established standard curves with known DNA quantities, e.g., 0.005 ng, 0.05 ng, 0.5 ng, 5 ng, 50 ng.

### 2.9. Statistical Analysis 

Experimental data analysis and graphical display of data were performed using statistical software GraphPad Prism 8. A comparison among the groups was assessed using the ANOVA test. The threshold for statistical significance was accepted as *p* < 0.05. Data are presented as mean ± SD, and each result represents a replicate of 3 independent experiments (n = 3).

## 3. Results

### 3.1. Characterization of Monolayer Differentiated ESCs

The cell viability over 7 days was optimal with a seeding density of 0.3 × 10^5^ cells/mL (Appendix A). No noticeable differences in the proliferative rate were observed in routine culture between alternate conditions. The expression of pluripotency markers (alkaline phosphatase, SSEA-1 (associated with loss of pluripotency), SSEA-4, Oct-3/4, and Nanog) was determined in undifferentiated and differentiated hESCs to confirm the retention of three-germ-layer differentiation and determine whether physoxia modulated expression levels. Consistent with established hESC behaviors, we noted a gradual decrease in the expression of alkaline phosphatase, SSEA-4, Oct-3/4, and Nanog, and increased SSEA-1 expression as the hESC populations became increasingly differentiated (Appendix A).

Three germ layer differentiation markers were explored to understand the spontaneous differentiation pattern by qRT-PCR of NES (Nestin) (ectoderm), TBXT (T-Box Transcription Factor T) (mesoderm), KDR (Kinase Insert Domain Receptor) (mesoderm), and AFP (Alpha-fetoprotein) (endoderm). Overall, we noted the increased expression of mesodermal markers in monolayer differentiated hESCs. 

The aggregated hESC data indicated substantial variation in NES expression as differentiation progressed, which nevertheless achieved levels of significance at days 5, 10, 20, and 40 in all conditions when compared to the undifferentiated cells (*p* < 0.01) (Figure 1). The differentiated SHEF1 cells displayed higher NES expression at day 5 in 21% AO versus 2% PG and 2% WKS (*p* < 0.001). The NES expression was subsequently downregulated on days 10 (*p* < 0.001) and 20 in all conditions (*p* < 0.001) versus the undifferentiated cells (Appendix A). The NES expression was higher in 21% AO versus 2% WKS in the undifferentiated SHEF2 cells (*p* < 0.01). Reduced NES expression was noted in the differentiated SHEF2 cells at day 5 in 2% PG (*p* < 0.05) and days 10, 20 in 2% WKS (*p* < 0.05) versus 21% AO. A significant decrease in the NES expression was noted for days 20 and 40 (*p* < 0.001) in all conditions compared to the undifferentiated cells (Appendix A).

Aggregated hESCs data indicated a significant decrease in KDR expression in 2% WKS compared to 21% AO in undifferentiated cells (*p* < 0.05). Overall, increased KDR expression was noted for all conditions during differentiation at days 10, 20, and 40 vs. undifferentiated cells (*p* < 0.01) (Figure 1). KDR expression was reduced in differentiated hESCs at day 5 in 2% PG and 2% WKS (*p* < 0.001) and day 20 in 2% WKS (*p* < 0.01) versus 21% AO. Increased KDR gene expression was observed in the majority of the monolayer differentiated cells at days 5, 10, 20, and 40 in all conditions compared to the undifferentiated SHEF1 cells (*p* < 0.01). The KDR expression was decreased at day 5 in 2% PG (*p* < 0.05) and 2% WKS (*p* < 0.01) versus 21% AO (Appendix A). Consistent with the above, a significant increase in KDR was also noted on days 5, 10, 20, and 40 in all conditions compared to the undifferentiated SHEF2 cells (*p* < 0.001). The undifferentiated and day 5 differentiated SHEF2 cells in 2% PG displayed reduced KDR expression when compared to air oxygen (*p* < 0.05). The reduced expression of KDR was observed in 2% WKS at days 10 and 20 compared to 21% AO (*p* < 0.05) (Appendix A).

Aggregated hESCs data reinforced the observation that undifferentiated (*p* < 0.01) and day 20 (*p* < 0.05) differentiated hESCs displayed reduced TBXT expression in physoxia versus AO. Consistent with the above, TBXT expression was elevated at differentiation days 5, 10, 20, and 40 vs undifferentiated (*p* < 0.001) (Figure 1). Elevated mesodermal marker, TBXT, expression was noted in the undifferentiated 21% AO SHEF1 and SHEF2 cells vs. 2% PG (*p* < 0.05) and 2% WKS. The TBXT expression was significantly elevated in the differentiated SHEF1 and SHEF2 cells at days 5, 10, 20, and 40 vs. the undifferentiated cells (*p* < 0.001) (Appendix A). The differentiated SHEF2 cells displayed reduced upregulation at day 20 in 2% PG and 2% WKS (*p* < 0.01) and again at day 40 in 2% PG (*p* < 0.01) when compared to AO (Appendix A).

Aggregated data showed decreased relative AFP expression at day 5, day 10, 20, and 40 of differentiation in all conditions compared to undifferentiated hESCs (*p* < 0.01) (Figure 1). The AFP expression in the monolayer differentiated SHEF1 cells was downregulated at day 10 in all conditions and, while undergoing some increase thereafter, failed in all but 2% WKS to recover to undifferentiated levels of expression (Appendix A). The undifferentiated SHEF2 cells in 2% PG and 2% WKS displayed reduced expression vs. 21% AO. The expression levels thereafter during differentiation displayed some evidence of fluctuation with an overall gradual decrease through to day 40 (Appendix A).

### 3.2. Differentiation-Driven Changes in TERT, Telomerase Activity, and Telomere Length Are Reduced in Physoxia

The TERT expression, telomerase activity, and telomere length were determined in the undifferentiated and differentiated hESCs. 

TERT expression. The TERT expression was higher in the undifferentiated SHEF1, SHEF2, and aggregated hESC data in physoxia vs. 21% AO (*p* < 0.01) (Figure 2 andAppendix A). The differentiated SHEF1 cells displayed progressively reduced TERT at differentiation days 5 (−1.01), 10 (−2.3), 20 (−3.6), and 40 (−7.3) in comparison to the undifferentiated SHEF1 cells in 21% AO (*p* < 0.001). Elevated TERT expression was noted at days 5 (−0.4), 10 (−1.4), and 20 (−3.0) in 2% PG vs. 21% AO (*p* < 0.05). Consistent with the above, differentiation days 5 (−0.2), 10 (−1.2), 20 (−1.5), and 40 (−5.4) in 2% WKS displayed higher TERT vs. 21% AO (*p* < 0.01) (Appendix A). 

Similarly, the SHEF2 cells displayed progressively reduced TERT at days 5 (−0.79), 10 (−2.2), 20 (−2.9), and 40 (−4.4) in 21% AO (*p* < 0.001) (Appendix A). The TERT expression was elevated in both 2% PG and 2% WKS in comparison to 21% AO after differentiation day 5, 10, 20, and 40 (2% WKS only) (*p* < 0.01). The aggregated hESC data was broadly reflective of the above, with elevated expression observed in the undifferentiated and differentiated populations cultured in physoxia vs. 21% AO (*p* < 0.05) (Figure 2). Confirmatory TERT protein expression levels were demonstrated via immunofluorescence (Appendix A).

Telomerase activity. The telomerase activity was elevated in the undifferentiated hESCs cultured in physoxia, achieving significance in the SHEF1 cells (*p* < 0.05). The aggregated ESC data indicated elevated telomerase activity in 2% PG and 2% WKS at days 5 (5.3 ± 0.3 and 5.5 ± 0.3, respectively), 10 (4.8 ± 0.5 and 4.9 ± 0.6, respectively), and 20 (3.8 ± 0.6 and 4.3 ± 0.4, respectively) compared to AO (4.4 ± 0.2, 3.7 ± 0.2 and 3.3 ± 0.2, respectively) (*p* < 0.01) (Figure 2). The telomerase activity in the SHEF1 cells followed the trend observed with TERT in displaying a progressive reduction as differentiation continued in 21% AO. Elevated telomerase activity was noted on days 5 and 10 in 2% PG (5.2 ± 0.1 and 4.5 ± 0.1, respectively) versus AO (*p* < 0.01). Similarly, higher telomerase expression was observed in the SHEF1 cells in 2% WKS (5.2 ± 0.3, *p* < 0.01, 4.1 ± 0.1, *p* < 0.05, and 4.0 ± 0.1, *p* < 0.001) compared to 21% AO on days 5, 10, and 20, respectively (Appendix A).

Elevated activity was also observed for the SHEF2 cells in 2% PG and 2% WKS at days 5 (5.4 ± 0.4 and 5.7 ± 0.2, respectively), 10 (4.9 ± 0.6 and 5.2 ± 0.3, respectively), 20 (4.2 ± 0.3 and 4.4 ± 0.4, respectively), and 40 (2% PG only, 3.5 ± 0.5) compared to AO (4.3 ± 0.1, 3.7 ± 0.2, 3.3 ± 0.3, respectively) (*p* < 0.01) (Appendix A). 

Telomere length. The hESCs displayed longer telomeres in physoxia overall, achieving significance for the SHEF2 cells in 2% WKS (*p* < 0.05). The pooled data showed a significant difference of telomere lengths in the undifferentiated stem cells cultured in 2% WKS (12.7 ± 1.3, *p* < 0.05) versus 21% AO (11.9 ± 1.2) and 2% PG (11.8 ± 1.0). Longer telomere lengths were noted in the undifferentiated cells vs. day 5 (11.0 ± 0.8, *p* < 0.05) and day 40 (8.4 ± 0.8, *p* < 0.01) cultured cells in 2% WKS compared to 21% AO. All the conditions demonstrated significant telomere shortening after days 5 (*p* < 0.001), 10 (*p* < 0.001) and 20 (*p* < 0.001). the telomere shortening rates were calculated as ~100 bp per doubling, irrespective of the culture condition applied, over 40 days differentiation (Appendix A). The undifferentiated SHEF1 cells displayed telomere lengths of 12.5 ± 1.1 kb (21% AO), 11.9 ± 1.1 kb (2% PG), and 12.9 ± 1.6 kb (2% WKS). Telomere shortening was consistent thereafter on days 5, 10, 20, and 40 (*p* < 0.001). The undifferentiated SHEF2 cells had significantly longer telomeres in the 2% WKS (12.5 ± 0.8, *p* < 0.05) versus 21% AO (11.2 ± 0.8) and 2% PG (11.6 ± 1.0) conditions. The telomere shortening was consistent across days 5 (*p* < 0.05), 10 (*p* < 0.001), 20 (*p* < 0.001), and 40 (*p* < 0.001). 

### 3.3. Physoxia Promotes Reduced TERT Promoter Methylation

We have previously detailed an overall reduction in global methylation in pluripotent stem cells cultured in physoxia in comparison to air oxygen [46]. We hypothesized that the TERT promoter would reflect these general observations and potentially reveal a specificity in their marking across regions. The TERT promoter regions were subsequently analyzed to determine the extent of their CpG methylation, including regions I (−1456, −1495 bp from TSS), II (−674, −717 bp from TSS), III (−315, −348 bp from TSS), IV (−122, −171 bp from TSS), and V (−67, −106 bp from TSS) in the undifferentiated and differentiated hESCs. Region I was highly methylated in all the undifferentiated hESCs and during differentiation, irrespective of the culture condition, indicating that it was responsive to neither differentiation nor oxygen and unlikely to play a role in the modulation of TERT expression. The aggregated hESC data indicated that the physoxic culture resulted in reduced CpG methylation in the undifferentiated cells across Regions II, III, IV, and V (Figure 3). Thereafter, and during differentiation, increased methylation was noted in both AO and 2% WKS for all Regions except I, achieving significance at Day 20 in Region V.

The undifferentiated SHEF1 cells displayed reduced CpG methylation in regions II, IV, and V in physoxic culture but not to a significant level in III (Appendix A Region II displayed progressively increased methylation levels through to differentiation day 20, which either stabilized or reduced thereafter but were significantly reduced at days 20 and 40 in comparison to 21% AO. Region III displayed culture-condition-independent increases in methylation through to day 20 while 2% WKS was reduced vs. 21% AO thereafter. Similarly, Region IV displayed a progressive increase in methylation levels across all the differentiation timepoints. Finally, Region V displayed markedly reduced methylation at day 20 in 2% WKS vs. 21% AO.

Consistent with the above, the undifferentiated SHEF2 cells displayed a significant reduction in methylation across regions II, III, IV, and V. Differentiation resulted in progressively increased methylation in regions II, IV, and V with no significant difference noted between 21% AO and 2% WKS (Appendix A). Region III displayed reduced levels of CpG methylation during differentiation at all timepoints, achieving significance at days 10 and 20 in 2% WKS vs. 21% AO. 

### 3.4. DNMT3B Expression Is Downregulated during Differentiation and Accelerated in Physoxia 

Having determined that the TERT promoter displayed reduced CpG methylation in the undifferentiated and differentiated hESCs cultured in physoxia, we next sought to determine if this was accompanied by the reduced expression of DNMT family members; DNMT1, DNMT3A, and DNMT3B. In the undifferentiated hESCs, we saw no significant differences in the DNMT1 expression levels, while both DNMT3A and DNMT3B displayed reduced expression, significantly so in the SHEF2 cells (Appendix A). DNMT1 expression was generally reduced as differentiation progressed with variable patterns or significant change between air- and physoxia-cultured hESCs (Appendix A and Figure 4). The DNMT3A expression in the SHEF1 cells was relatively stable through to day 10 while the SHEF 2 cells displayed a consistent and progressive decline throughout the differentiation time course that was mirrored in the overall hESC expression dynamic (Appendix A and Figure 4). Responses to oxygen were variable and cell line-specific. Finally, DNMT3B was consistent in its progressively reduced expression across 40 days of differentiation in both the SHEF1 and SHEF2 cells, significantly reduced vs. air oxygen at days 5 and 10 (SHEF1) 20 and 40 (SHEF 2), and days 5, 10, 20, and 40 in the pooled hESCs (Appendix A and Figure 4). 

### 3.5. NA Treatment Decreased DNMT3B Gene Expression and Enzyme Activity

A DNMT3B-selective inhibitor, NA, was used to evaluate the impact of reduced DNMT3B activity on the methylation and epigenetic regulation of TERT. We first exposed the undifferentiated and differentiated hESCs (SHEF2) to an NA dose–response curve using a WST1 assay as an output of the maintenance of metabolic viability. We identified significant divergence from the DMSO-treated controls at concentrations above 40 nM where the hESCs cultured in 2% WKS were apparently more sensitive than those cultured in air (Figure 5A). The differentiated cells displayed a higher tolerance to NA with a maximum non-toxic drug dose determined at 310 nM (Figure 5B). We next evaluated the impact of the DNMT3B activity inhibitor, NA, on the DNMT3B gene expression in the undifferentiated and differentiated SHEF2 cells. The undifferentiated cells displayed significantly decreased DNMT3B expression after NA treatment in 21% AO (−3.3, *p* < 0.001) and 2% WKS (−3.5, *p* < 0.001) (Figure 5C). Decreased expression was also associated with NA supplementation at day 5 in 21% AO (*p* < 0.001) and 2% WKS (*p* < 0.01). In contrast, as differentiation progressed, the NA-treated cells displayed elevated expression at days 20 and 40 in 21% AO (*p* < 0.01) and 2%WKS (*p* < 0.01) in comparison to the untreated cells (Figure 5C). Finally, we confirmed that NA supplementation inhibited DNMT3B enzyme activity in both the undifferentiated and differentiated hESCs, using the previously determined dose regime. Consistent with our transcriptional observations (Figure 5C), we observed a progressive and significant reduction in the DNMT3B activity in the differentiating SHEF2 cells. The NA supplementation promoted a further reduction in DNMT3B activity at all timepoints tested (Figure 5D). 

### 3.6. Inhibition of DNMT3B Increased TERT Expression and Activity during Differentiation

Having determined that a non-toxic dose of supplementation inhibited DNMT3B transcriptional downregulation and promoted enzymatic inhibition in the differentiated SHEF 2 cells, we next sought to determine the resultant impact on the TERT expression and activity. For experimental clarity, we only explored the physoxia cultures, removing air or physiological hyperoxia the subsequent analyses. Consistent with our earlier demonstrations that TERT transcriptional downregulation was associated with increased promoter methylation (Figure 3), we observed reduced transcriptional repression from day 10 onwards in the NA-supplemented SHEF2 cells vs. 2% WKS alone (*p* < 0.001) (Figure 6A). Further, while no significant differences were noted for telomerase activity in the NA-supplemented SHEF 2 cells through to day 20, we did observe significantly higher telomerase activity (4.22 ± 0.19, *p* < 0.001) after 40 days of the NA treatment in comparison to the untreated cells (2.84 ± 0.29) (Figure 6B). 

### 3.7. DNMT3B Inhibition Reduced TERT Promoter Methylation 

We reasoned that increased TERT expression and telomerase activity should be associated with decreased TERT promoter methylation. To determine this, we repeated our pyrosequencing analysis of the TERT promoter regions with or without NA supplementation (Figure 7). Region I, as described previously, displayed no change in the methylation level during differentiation, while the NA supplementation induced a ~25% reduction in the undifferentiated and differentiated SHEF2 populations. Regions II and III displayed a sharp reduction in methylation levels, ~50–75%, when exposed to NA, indicating a dynamic methylation signature and a key role for DNMT3B in its regulation. In contrast, Regions IV and V displayed little response to the NA supplementation with their overall low levels of methylation unchanged.

### 3.8. DNMT3B Association with TERT Promoter Increases during Differentiation

We last sought to confirm that DNMT3B associated directly with the TERT promoter and determine if the extent of its association reflected the methylation behaviors observed previously. The DNMT3B-TERT ChIP confirmed binding across Regions II to V of the TERT promoter (Figure 8). Consistent with our earlier observations, we noted that DNMT3B binding to Region II increased in the differentiated SHEF2 cells on days 10 (3.1, *p* < 0.01), 20 (3.5, *p* < 0.001), and 40 (3.9, *p* < 0.001) when compared to the undifferentiated SHEF2 cells (2.5). Further, elevated DNMT3B binding on Region III was noted after 20 (4.6, *p* < 0.001) and 40 days (4.9, *p* < 0.001) in the differentiated vs. undifferentiated cells (3.7). Similarly, Region IV had elevated DNMT3B binding after 20 (3.0, *p* < 0.001) and 40 days (3.1, *p* < 0.001) vs. the undifferentiated cells (2.5). Region V also displayed increased DNMT3B binding at days 10 (2.5, *p* < 0.01), 20 (2.7, *p* < 0.001), and 40 (2.8, *p* < 0.001) in 2% when compared to the undifferentiated cells (2.1).

## 4. Discussion

With an unlimited proliferative potential and ability to differentiate into a wide range of cell types, pluripotent stem cells have the potential to produce human tissues, treat genetic illnesses, and be utilized across a broad range of developmental and fundamental biology research [43,44]. Pluripotent ESCs’ hallmark characteristics include high proliferation, colony formation, Oct-4 and Nanog expression, alkaline phosphatase activity, telomerase activity, and long telomeres [1]. However, the telomerase activity and TERT gene expression are reduced during differentiation, indicating that telomerase regulation plays an important role in embryonic development [47,48,49,50]. In this study, we have shown that telomerase regulation is oxygen sensitive during differentiation and epigenetically regulated by the DNMT3B enzyme.

ESCs and iPSCs lose pluripotency marker expression as they differentiate [51,52,53]. We observed that ESC monolayer differentiation is accompanied by the downregulation of pluripotency genes (Appendix A). ESCs, isolated from pre-implantation blastocysts, can differentiate into all embryonic lineages [54] in vivo and in vitro where 3D aggregates and embryoid bodies express upregulated Nestin (ectodermal) [55], Brachyury (mesodermal), and AFP (endodermal) markers [56]. Similar to previous studies, we observed the increased expression of differentiation markers after the monolayer differentiation of stem cells in vitro and the increased expression of mesodermal markers. Three germ layer differentiation markers were used to assess the ESCs’ monolayer differentiation status. The monolayer differentiated ESCs displayed retentions of, but overall reductions in, ectodermal (NES) or endodermal (AFP) marker expression, in comparison to the undifferentiated hESCs. There was an increased expression of mesodermal markers (TBXT and KDR) in the hESCs during differentiation, with significantly less in physiological oxygen culture than AO. 

The decreased expression of TERT in the differentiated ESCs is described as being associated with decreased proliferation, increased G1 phase, the promotion of differentiation, and a failure to produce stable ESC sublines [10,47,48,49,50,57]. Consistent with previous literature, the TERT expression, telomerase activity, and TERT protein levels (via immunofluorescence) displayed a gradual downregulation during 40 days of differentiation in the ESCs. The TERT, telomerase, and telomere lengths were considerably higher in the ESCs cultivated in physiological oxygen niches. Taken together, these highlight an oxygen-linked sensitivity for multiple components of telomere and telomerase biology. 

The physiological oxygen environment is a critical component of stem cell biology and cell fate [38,58]. Early embryos exist in a reduced oxygen environment due to the absence of vascularization in early development, and ESCs are optimally maintained in physiological normoxia (2–5% O_2_ [38,58,59,60,61]. We observed that reduced oxygen culture (2% PG and 2% WKS) increased the proliferation rate of the ESCs in comparison to AO (Appendix A). Many researchers have explored epigenetic mechanisms including DNA methylation and histone changes in cellular processes, such as differentiation and embryonic developmental programming [21,28]. DNA methylation and hydroxymethylation have essential roles in PSC functions, differentiation, cell fate, and the maintenance of characteristics in low oxygen environments [21,62,63,64]. DNMT1 is a maintenance methylase during replication with an affinity for hemimethylated DNA. DNMT3A and DNMT3B collaborate with DNMT1 to perform de novo methylation, including during early development [65]. DNMT3A and DNMT3B are highly expressed in undifferentiated ESCs [23,66,67], but downregulated during differentiation [22]. We observed a significant reduction in the expression of maintenance and de novo methyltransferases DNMT1, DNMT3B, and DNMT3A during the differentiation of the ESCs. Our results showed a sharp decrease in the expression of DNMT3B during pluripotent stem cell differentiation, in line with previous studies [68].

The methylation alteration of the TERT promoter is strongly linked to its regulation [69]. ES and somatic cell methylation are distinct and distinguishes pluripotent stem from somatic cells [21]. Previous data has showed a lower methylation signature in ESCs, similar to our observations [70,71,72]. We observed that the distal promoter (region I (−1456, −1495 bp from TSS)) has a high methylation level in all conditions with no significant difference during differentiation. Takasawa et al. demonstrated that the highly methylated TERT region (−845, −807 from TSS) upregulated promoter activity in iPSCs compared to their parental somatic cells, indicating further complexity in promoter regulation [73]. The methylation status on proximal promoter regions II, III, IV, and V was more responsive to change, with emphasis on regions II and III. Our data showed a strong link between telomerase activity and oxygen-sensitive TERT proximal promoter methylation. There was a substantial increase in methylation levels during differentiation. 

Molecular docking studies of DNMT3B suggest that NA forms hydrogen bonds with Glu697, Arg731, Arg733, and Arg832 for the stabilization of the protein–ligand complex and selectively inhibits DNMT3B [74,75]. NA (5 µmol/L) induced no changes in the transcript levels of DNMT3B or DNMT1, but selectively inhibited DNMT3B activity [73]. We investigated the effect of DNMT3B inhibition on TERT gene expression and telomerase activity. Interestingly, we observed an increase in TERT gene expression after 20 days of differentiation and in telomerase activity after 40 days of differentiation compared to the untreated cells. Pyrosequencing indicated that DNMT3B inhibition reduced TERT promoter methylation, and was associated with increased gene and enzyme activity. The ChIP-qPCR data revealed that DNMT3B binding to the TERT promoter was correlated with increased methylation during differentiation and was associated with decreased TERT gene and telomerase activity.

In conclusion, understanding the mechanism behind the reversible silencing of the TERT gene during differentiation, embryonic development, and aging, or conversely, its activation in cancer, has potential for informing future clinical applications, cancer treatment, diagnosis, prognosis, and cellular aging research. We have highlighted the link between TERT promoter methylation, TERT expression, and correlated this to telomerase and the DNMT3B enzyme. We suggest that the proximal promoter region is a potential target to regulate TERT expression and modify telomerase activity. The careful application of stem cell models can contribute to the development of useful epigenetic engineering tools for brand-new clinical applications.

## Figures and Tables

**Figure 1 biomolecules-14-00131-f001:**
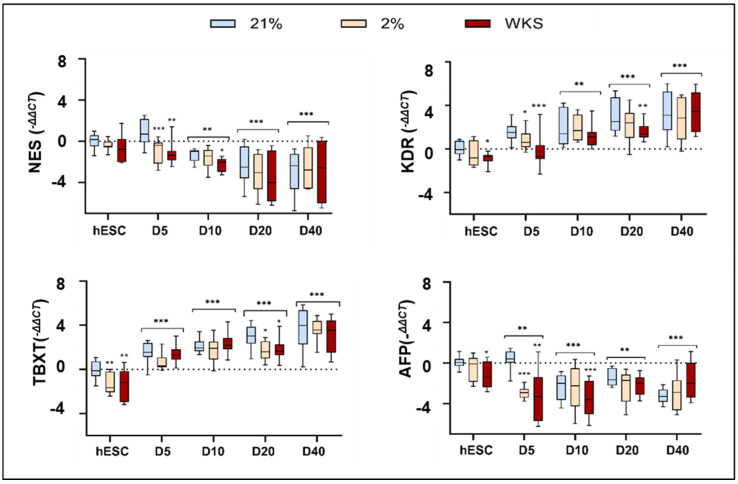
Increased mesodermal marker expression in monolayer differentiated hESCs. Expression of three germ layer (Ectoderm (NES), Mesoderm (KDR, TBXT), and Endoderm (AFP)) differentiation markers was analyzed in pooled hESCs data in 21% AO, 2% PG, and 2% WKS. The *y*-axis indicates expression (−ΔΔCT) is normalized to GAPDH. Data are represented as n = 3 × 3 (three biological samples and each sample has three replicates), * *p* < 0.05, ** *p* < 0.01, *** *p* < 0.001 vs. undifferentiated counterpart where connecting line is present, otherwise is indicative of vs. 21% O_2_. *x*-axis; D indicates day of differentiation.

**Figure 2 biomolecules-14-00131-f002:**
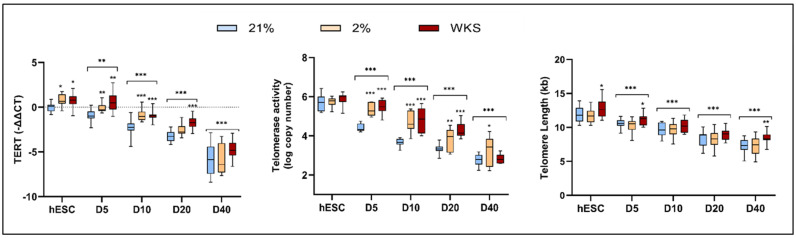
Physoxia slows downregulation of TERT, telomerase activity, and rate of telomere shortening in differentiating hESCs cultured in air oxygen (21% AO) and physiological oxygen conditions (2% PG and 2% WKS) are presented for TERT expression, telomerase activity, and telomere shortening across 40 days differentiation. Data are presented as mean ± standard deviation (SD), n = 3 × 3, * *p* < 0.05, ** *p* < 0.01, *** *p* < 0.001 vs. undifferentiated counterpart where connecting line is present, otherwise is indicative of vs. 21%O_2_. *x*-axis; D indicates day of differentiation.

**Figure 3 biomolecules-14-00131-f003:**
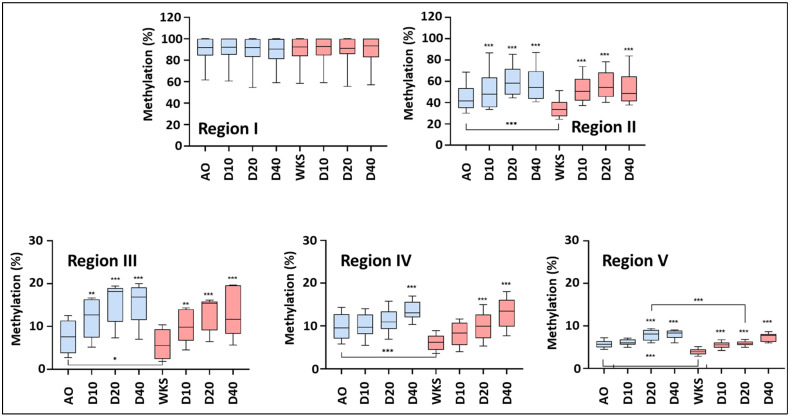
Promoter-region-specific methylation of TERT promoter in hESCs. Promoter regions (**I**–**V**) relative to TSS were evaluated using pyrosequencing in air oxygen (AO) (light blue bars) and 2% O_2_ WKS (WKS) (pink bars). *y*-axis indicates DNA methylation level (%) at CpG sites, and *x*-axis indicates sample identity (D indicates day of differentiation) and timepoint of differentiation. Data presented as median (min-max). n = 3, * *p* < 0.05, ** *p* < 0.01, *** *p* < 0.001 vs. undifferentiated counterpart or as indicated by connecting line.

**Figure 4 biomolecules-14-00131-f004:**
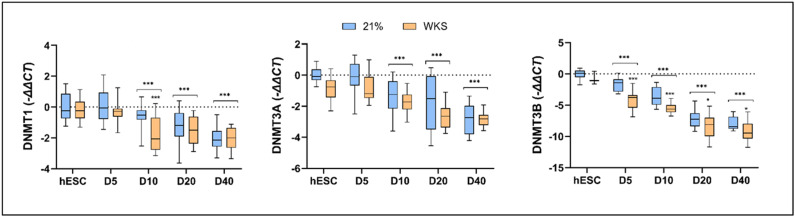
DNMT family members display distinct downregulation profiles during hESC differentiation. DNMT1, DNMT3A, and DNMT3B expression data are presented for pooled undifferentiated and differentiated (D) hESCs in 21% AO and 2% WKS. DNMT expression (−ΔΔCT) is normalized to the expression of GAPDH. Data are represented as mean ± standard deviation (SD), n = 3, * *p* < 0.05, *** *p* < 0.001 vs. undifferentiated counterpart where connecting line is present, otherwise is indicative of vs. 21% O_2_.

**Figure 5 biomolecules-14-00131-f005:**
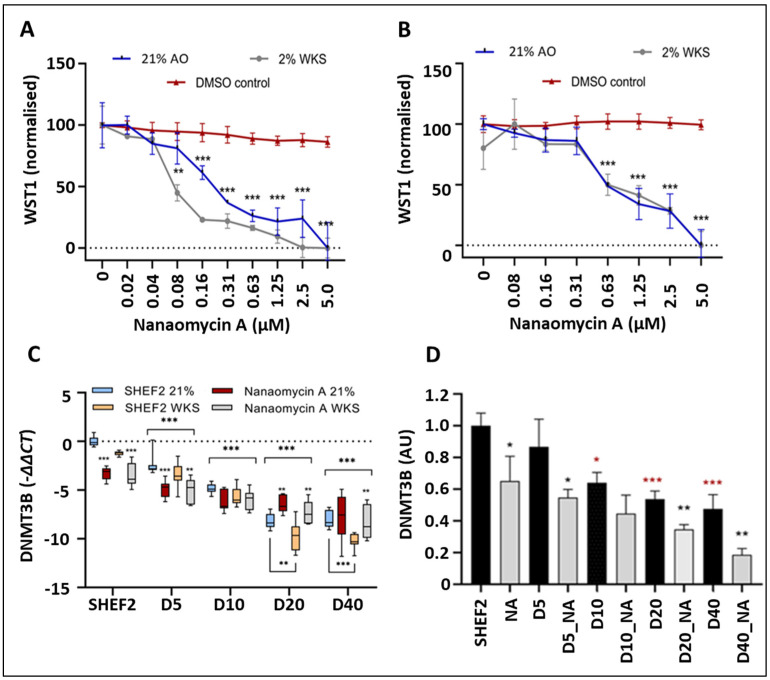
DNMT3B activity, but not expression, is downregulated by NA supplementation. Undifferentiated (**A**) and differentiated SHEF2 cells (**B**) were treated with increasing doses of NA ranging from 20 nM to 5 µM for seven days, in both AO and WKS. Cell viability (WST1 (normalized)) was plotted against NA concentrations in µM. (**C**) DNMT3B expression in differentiating SHEF2 across a 40-day timeline; with and without NA supplementation. Data normalized to GAPDH expression. (**D**) DNMT3B enzyme activity across a 40-day differentiation timeline, with and without NA supplementation. Data are represented as mean ± standard deviation (SD), n = 3, * *p* < 0.05, ** *p* < 0.01, *** *p* < 0.001 vs. DMSO control (**A**,**B**), vs. undifferentiated counterpart where connecting line is present or as indicative or vs. unsupplemented comparator (**C**), or unsupplemented comparator (black asterisk) or undifferentiated control (red asterisk).

**Figure 6 biomolecules-14-00131-f006:**
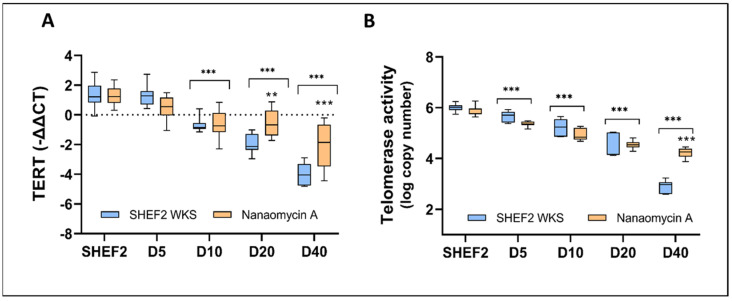
DNMT3B enzyme inhibition slows TERT downregulation and elevates telomerase activity at late-stage differentiation. (**A**) TERT expression in differentiating SHEF2 cells across a 40-day timeline; with and without NA supplementation. Data normalized to GAPDH expression. (**B**) Telomerase activity in differentiating SHEF2 cells across a 40-day timeline; with and without NA supplementation. Data are represented as mean ± standard deviation (SD), n = 3, ** *p* < 0.01, *** *p* < 0.001 vs. undifferentiated counterpart where connecting line is present, otherwise is indicative of vs. unsupplemented comparator.

**Figure 7 biomolecules-14-00131-f007:**
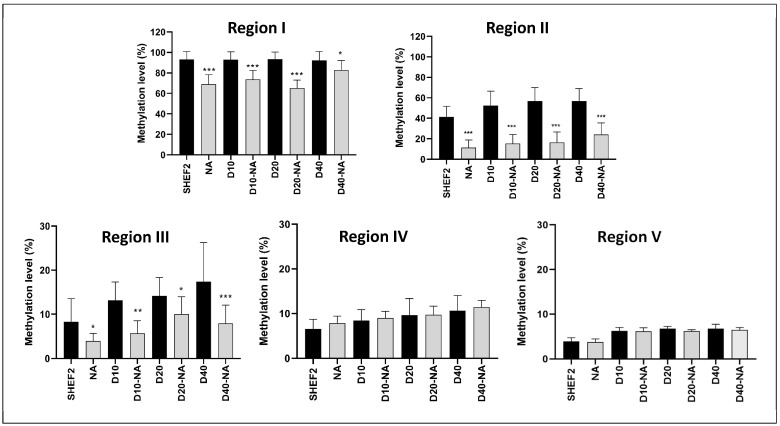
DNMT3B inhibition reveals dynamic TERT promoter methylation. Promoter regions (**I**–**V**) relative to TSS were evaluated using pyrosequencing in 2% O_2_ WKS with and without NA supplementation. *y*-axis indicates DNA methylation level (%) at CpG sites, and *x*-axis indicates sample identity; D indicates day of differentiation and timepoint of differentiation. Data presented as median (min-max). n = 3, * *p* < 0.05, ** *p* < 0.01, *** *p* < 0.001 vs. unsupplemented comparator.

**Figure 8 biomolecules-14-00131-f008:**
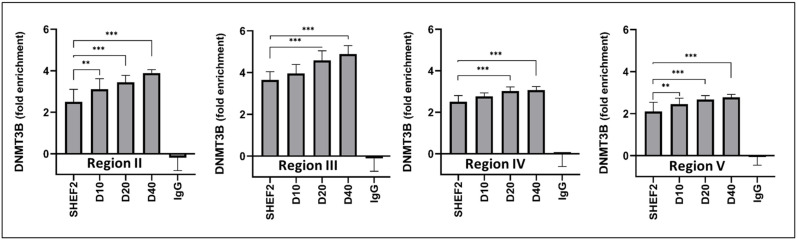
DNMT3B association with TERT promoter increases during differentiation. DNMT3B enzyme association with TERT promoter regions was established using ChIP qPCR. Data are represented as mean ± standard deviation (SD), n = 3, ** *p* < 0.01, *** *p* < 0.001 as indicated via connecting lines. CpG methylation regions II (−674, −717 bp from TSS), III (−315, −348 bp from TSS), IV (−122, −171 bp from TSS), and V (−67, −106 bp from TSS). D indicates day of differentiation and timepoint of differentiation.

## Data Availability

Data are available in a publicly accessible repository.

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
