# Peer review of "TERT Promoter Methylation Is Oxygen-Sensitive and Regulates Telomerase Activity"

_biomolecules, 2024, doi:10.3390/biom14010131_

Round 1

Reviewer 1 Report

Comments and Suggestions for Authors

The authors studied the effect of oxygen levels in cultured cells and the effect on TERT promoter methylation. Across the manuscript, a large amount of RT-PCR data is presented. The way the manuscript is presented is difficult to know the questions they want to answer. For instance, having 10 Figures doesn’t help in the flow of the paper. 

Here are my suggestions:

The authors start the result section with a series of RT-PCR data from a variety of genes to describe differentiation in ESCs, however, it is not clear what question they want to answer. A different way to introduce this section would make readers understand better their objectives. 

The authors showed in Figure 3 an increase in methylation particularly in Region III and Region IV and in Region III of Figure 4 of the TERT promoter, how to conciliate that DNMT3B association with TERT promoter increases during differentiation (Figure 10) but TERT expression decreases? This point should be discussed. 

The methylation levels of TERT promoter (Figure 4) region III are very different to the one shown in Figure 9 (black bars), why is there such a big discrepancy? 

Too many graphs in the main text which render it confusing and heavy to read, for instance, Figure 3, 4 and 5 show basically the same results, I suggest keeping only Figure 5 in the main text and moving 3 and 4 as supplementary information. 

Reference to add:

Takasawa, K., Arai, Y., Yamazaki-Inoue, M. et al. DNA hypermethylation enhanced telomerase reverse transcriptase expression in human-induced pluripotent stem cells.Human Cell 31, 78–86 (2018). https://doi.org/10.1007/s13577-017-0190-x

Other points:

The abstract is very long given details that are not relevant at that stage of the paper, e.g. "....oxygen workstation to provide a fully defined 2% O2 environment." For instance, what is the difference between the O2 incubator used and the O2 workstation? No details are given in the material and methods section. In summary, the abstract needs to be more concise, presenting the relevance of the work. 

The colour code in Figure 2 and Figure 3 is not given.

The stars showing significance in all graphs are sometimes confusing, eg. In Figure 8A D20, the 3 stars are the comparison between the blue and yellow bars, but the 2 stars are not clear what are referring to. This applies to many of the graphs. 

In many cases, the error bars in the graphs are big, but still, three stars for significance are given, Did the authors test for normality in their data? Is ANOVA adequate for this kind of analysis? it seems that having three replicates as indicated in the material and methods section, a non-parametric statistical test will be more adequate. In addition, in Figure 8B, it is difficult to imagine 3 starts for condition D20, where the standard deviation of the blue bar completely covers the yellow one, and that is the case for other conditions such as Figure 8A D10 for example. 

Author Response

Dear MDPI Editorial Office,

We are grateful to MDPI for undertaking such a thorough and constructive review of our manuscript. Further, we welcome the opportunity to address the concerns collectively highlighted by the reviewers and drawn together below with our response and resultant revisions highlighted.

- Reviewer 1

Point 1) The authors start the result section with a series of RT-PCR data from a variety of genes to describe differentiation in ESCs, however, it is not clear what question they want to answer. A different way to introduce this section would make readers understand better their objectives.

Response 1) We have added additional explanation into page 6. Three germ layer differentiation markers were explored to understand the pattern of spontaneous differentiation and determine if physoxia played an influencing role. Overall, we noted an increased expression of mesodermal markers in monolayer differentiated hESC. (Page 6, lines 225-239; clean).

For clarity we have moved SHEF1 and SHEF2 Figures into the supplementary section and retained Figures detailing aggregated hESCs data within the manuscript.

Point 2) The authors showed in Figure 3 an increase in methylation particularly in Region III and Region IV and in Region III of Figure 4 of the TERT promoter, how to conciliate that DNMT3B association with TERT promoter increases during differentiation (Figure 10) but TERT expression decreases? This point should be discussed.

Response 2). High methylation at the promoter region often leads to gene silencing. This is because methylated DNA is less accessible to the transcriptional machinery, making it difficult for the cellular machinery to initiate transcription.

In summary, there was a substantial increase in TERT promoter methylation levels during differentiation. DNMT3B binding to TERT promoter was correlated with increased methylation during differentiation and was associated with decreased TERT gene and telomerase activity. See page 14, line 534 – 556 for discussion; clean.

Point 3) The methylation levels of TERT promoter (Figure 4) region III are very different to the one shown in Figure 9 (black bars), why is there such a big discrepancy?

Response 3) The methylation levels of TERT promoter in Figure 4 (now in Supplementary following advice from Point 4 below) and Figure 9 (now Figure 7) reflect independent experimentation. Figure 9 (Figure 7) details the impact of DNMT3B inhibition on TERT methylation over time. The unsupplemented SHEF2 follow broadly the same pattern as previous experimentation demonstrated (Figure 4 now in Supplementary).

Point 4) Too many graphs in the main text which render it confusing and heavy to read, for instance, Figure 3, 4 and 5 show basically the same results, I suggest keeping only Figure 5 in the main text and moving 3 and 4 as supplementary information.

Response 4) There was a general reviewer`s point on graphs and we agree with reviewers. We have moved figures 3 and 4 from manuscript into supplementary figures.

Point 5) Reference to add:

Takasawa, K., Arai, Y., Yamazaki-Inoue, M. et al. DNA hypermethylation enhanced telomerase reverse transcriptase expression in human-induced pluripotent stem cells.Human Cell 31, 78–86 (2018). https://doi.org/10.1007/s13577-017-0190-x

Response 5) We appreciate the contribution. We have added the reference to our discussion. Page 14, lines 539 – 541; clean.

Point 6) The abstract is very long given details that are not relevant at that stage of the paper, e.g. "....oxygen workstation to provide a fully defined 2% O2 environment." For instance, what is the difference between the O2 incubator used and the O2 workstation? No details are given in the material and methods section. In summary, the abstract needs to be more concise, presenting the relevance of the work.

Response 6) We have removed information from the abstract and placed it into the materials and methods section (Page 3, lines 113-118; clean).

`Cells were maintained in either air oxygen (21% AO), a fully defined 2% O2 environment (workstation) (2% WKS), and a standard 2% O2 incubator (2% PG) where samples were handled in a standard class II biological safety laminar flow cabinet. Media was deoxygenated to a 2% O2 level using defined Hypoxycool (Baker Ruskinn, Bridgend, UK) cycle settings. Pre-gassed media (pre-conditioned to 2% O2 in a HypoxyCool unit) was used in all 2% O2 experimentation. `

Point 7) The colour code in Figure 2 and Figure 3 is not given.

The stars showing significance in all graphs are sometimes confusing, eg. In Figure 8A D20, the 3 stars are the comparison between the blue and yellow bars, but the 2 stars are not clear what are referring to. This applies to many of the graphs.

Response 7) Figures 2 and 3 have been revised and now include a figure-based (Figure 2) or legend-based guide.

For clarity the connecting line and asterisk(s) indicates significance vs. undifferentiated counterparts. Where unaccompanied by a connecting line the asterisk(s) indicates significance vs. air oxygen unless otherwise stated. Legends have been amended to reflect this.

Point 8) In many cases, the error bars in the graphs are big, but still, three stars for significance are given, Did the authors test for normality in their data? Is ANOVA adequate for this kind of analysis? it seems that having three replicates as indicated in the material and methods section, a non-parametric statistical test will be more adequate. In addition, in Figure 8B, it is difficult to imagine 3 starts for condition D20, where the standard deviation of the blue bar completely covers the yellow one, and that is the case for other conditions such as Figure 8A D10 for example.

Response 8) Our statistical advice indicated that while nonparametric tests don’t require data that are normally distributed they have the disadvantage that groups in a nonparametric analysis typically must all have the same variability (dispersion). Nonparametric analyses can then provide inaccurate results when variability differs between groups. Conversely, parametric analyses, like the multiple t-test or two-way ANOVA, allow you to analyze groups with unequal variances. This removes the potential confounder of differing variability between groups when parametric analysis is applied.

We are appreciative of the suggestion but for the reason given above are confident that two-way ANOVA is appropriate in this instance. For thoroughness we have also explored the application of non-parametric tests and see similar results to the two-way ANOVA with retention of significance

For clarity the connecting line and asterisk(s) indicates significance vs. undifferentiated counterparts. Where unaccompanied by a connecting line the asterisk(s) indicates significance vs. air oxygen unless otherwise stated. Legends have been amended to reflect this.

Reviewer 2 Report

Comments and Suggestions for Authors

The authors describe a very comprehensive study of the activity of the hTERT promoter of which methylation is highly oxygen dependent thus regulating subsequent telomerase activity.

I highly enjoyed reading this manuscript. However, the major question regarding it is its probable importance. I think that the authors should refer to it in a more expanded way in the discussion.

Apart from this point, the figures should include the explanation of the various colors used in the graphs..

Author Response

Dear MDPI Editorial Office,

We are grateful to MDPI for undertaking such a thorough and constructive review of our manuscript. Further, we welcome the opportunity to address the concerns collectively highlighted by the reviewers and drawn together below with our response and resultant revisions highlighted.

- Reviewer 2

Point 1) the major question regarding it is its probable importance. I think that the authors should refer to it in a more expanded way in the discussion.

Response 1) We have highlighted the importance in Page 14, lines 565-573.

`In conclusion, understanding the mechanism behind reversible silencing of the TERT gene during differentiation, embryonic development, and ageing, or conversely activation in cancer has potential for informing future clinical applications, cancer treatment, diagnosis, prognosis, and cellular ageing research. We have highlighted the link between TERT promoter methylation, TERT expression, and correlated this to telomerase and DNMT3B enzyme. We suggest that proximal promoter region is potential target to regulate TERT expression and modify telomerase activity. Careful application of stem cell models can contribute to the development of useful epigenetic engineering tools for brand new clinical applications.`

Point 2) Apart from this point, the figures should include the explanation of the various colors used in the graphs.

Response 2) Figures are updated and we included explanation of the bar colours.

Reviewer 3 Report

Comments and Suggestions for Authors

Epigenetic regulation of telomerase activity driven by oxygen levels is a valuable proposition.

Personally, I do not see any problems when researchers do not use cutting-edge methods. My major concern is striking inconsistencies in presented data. Why do the two ES cell lines used in this study show such different patterns during differentiation under specified oxygen conditions? I could not understand what significant differences are indicted by numerous stars. What n=3 or n=3x3 stands for is not clear. Were all experiments done in 3 biological replicates? Were all samples in each experiment grown and treated simultaneously? Why do SDs vary so much?

The fact that DNMT3B inhibition causes the reduction of DNA methylation in predictable. When TERT promotor methylation is reduced an increase in TERT expression is expected. How these changes in the promoter methylation affect telomerase activity is the question. Why do we see some sort of opposite effect (Fig. 8)? And where are the data @ 21% air oxygen conditions?

Comments on the Quality of English Language

The paper as a whole is poorly written (logic and style) and poorly edited (grammar, typos and punctuation).

Author Response

Dear MDPI Editorial Office,

We are grateful to MDPI for undertaking such a thorough and constructive review of our manuscript. Further, we welcome the opportunity to address the concerns collectively highlighted by the reviewers and drawn together below with our response and resultant revisions highlighted.

- Reviewer 3

Point 1) My major concern is striking inconsistencies in presented data. Why do the two ES cell lines used in this study show such different patterns during differentiation under specified oxygen conditions? I could not understand what significant differences are indicted by numerous stars. What n=3 or n=3x3 stands for is not clear. Were all experiments done in 3 biological replicates? Were all samples in each experiment grown and treated simultaneously? Why do SDs vary so much?

Response 1)

We understand that the presentation of the significant differences lack clarity. In the first instance, we have compared undifferentiated hESCs with their differentiated counterparts and indicated significance using asterisks with line connections indicating comparison. The second comparison, indicated by asterisks only is between oxygen conditions at individual time points.

n=3 indicates three biological samples. n=3x3 indicates three biological samples where each sample has three technical replicates. Samples were grown and treated simultaneously with consistent profiles emerging during differentiation between cell lines. The SD deviation variation is anticipated when we combined data from non-identical hESC lines. The reproducibility of overall trends, despite use of non-identical cell lines, adds credibility to our observations.

Point 2) The fact that DNMT3B inhibition causes the reduction of DNA methylation in predictable. When TERT promotor methylation is reduced an increase in TERT expression is expected. How these changes in the promoter methylation affect telomerase activity is the question. Why do we see some sort of opposite effect (Fig. 8)? And where are the data @ 21% air oxygen conditions?

Response 2) We agree with reviewer that the reduction of DNA methylation was predictable and that which nevertheless required demonstration and evidencing. Further, while reduced TERT promoter methylation and increased TERT expression is also expected this too requires demonstration and evidence. We see variability in promoter region methylation leading us to hypothesize that distinct promoter regions may play a stronger role in TERT transcriptional regulation than others.

We are unclear what the reviewer refers to when the cite ‘opposite effect’ in relation to Figure 8. Chemical inhibition of DNMT3B resulted in reduced transcriptional repression of TERT from D10 onwards but no significant change in either gene expression or telomerase activity in undifferentiated cells. DNMT3B inhibition was reflected in a significant increase in TERT expression and telomerase activity at late-stage differentiation (D40). Important to note is that we used lower, non-toxic, concentrations of Nanaomycin A for undifferentiated cells than with their differentiated counterparts which may have influenced results.

Point 3) The paper as a whole is poorly written (logic and style) and poorly edited (grammar, typos and punctuation).

Response 3) The manuscript has been edited throughout by a fluent, native, English speaker and experienced author. We are appreciative of the feedback and have re-reviewed throughout. We are unaware of where specific examples have altered interpretation or otherwise.

Round 2

Reviewer 1 Report

Comments and Suggestions for Authors

The authors have answered my concerns about the manuscript. 

Author Response

Thank you